# Exploring Cultural Bias in Two Different Motor Competence Test Batteries When Used in African Children

**DOI:** 10.3390/ijerph19116788

**Published:** 2022-06-01

**Authors:** Bouwien Smits-Engelsman, Evi Verbecque, Marisja Denysschen, Dané Coetzee

**Affiliations:** 1Department of Health & Rehabilitation Sciences, Faculty of Health Sciences, The University of Cape Town, Cape Town 7925, South Africa; 2Physical Activity, Sport and Recreation, Faculty Health Sciences, North-West University, Potchefstroom 2520, South Africa; marisja.denysschen@gmail.com (M.D.); dane.coetzee@nwu.ac.za (D.C.); 3Rehabilitation Research Centre (REVAL), Rehabilitation Sciences and Physiotherapy, Hasselt University, Agoralaan Building A, 3590 Diepenbeek, Belgium; evi.verbecque@uhasselt.be

**Keywords:** motor competence, African children, assessment, validity

## Abstract

Both the Movement Assessment Battery for Children second edition (M-ABC-2) and Bruininks-Oseretsky Test of Motor Proficiency second edition short form (BOT-2-SF) are frequently used in research and in the clinical practice to evaluate motor competence in children. Despite its widespread use in research, no studies have reported the results of case identification in African children. Comparing these two motor assessment tools for a different target group is important in order to select the most appropriate clinical and research tool. Methods. A total of 444 children performed MABC-2, 165 children also performed the BOT-2-SF and subsamples were tested on specific subtests of the BOT-2 (Running and Agility, Balance, and Strength). Tests were administered to randomly selected children between 6 and 10 years of age. Results: 36% for the children scored at or below the 16th percentile of the MABC-2, while this was 43%, 27%, and 23% for the component score in Manual Dexterity, Aiming and Catching, and Balance, respectively. Of the children 16% scored at or below the 17th percentile of the BOT-2-SF total score, while this was 3%, 9% and 22% for the subtest scores Running and agility, Balance, and Strength, respectively. A moderate correlation (r = 0.44) was found between total scores of the two tests. No significant correlations were found between the dynamic MABC-2 item (Jumping/Hopping) standard scores and any of the 9 balance items of the BOT-2. Conclusion: Far more children scored in the clinical “at risk” range (<16th percentile) when tested with the MABC-2 than with the BOT-2-SF. Overall, these children seemed not to be limited in motor performance measured by the BOT-2-SF, Running and Agility, and Balance. South African children did show lower levels of strength and explosive power. Children from different cultures will need tests for the specific motor skills that are representative for optimal functioning in their own setting. Thus, adapting reference norms and cut-off values may not be the optimal solution.

## 1. Introduction

Neurodevelopmental disorders such as Developmental Coordination Disorder (DCD) are under recognized in developing countries [1,2]. Current understanding about the causes of various neurodevelopmental disorders has largely relied on samples from European ancestry, creating concern about the relevance of the science for African populations and global health equity. To objectively measure motor competence, the child’s motor abilities need to be measured with a reliable and valid tool.

The most widely used motor tests to assist in the diagnosis of developmental motor delays are the Movement Assessment Battery for Children, second edition (MABC-2) and the Bruininks–Oseretsky Test of Motor Proficiency, second edition (BOT-2) [3,4] as recommended by the international guideline [5]. Both tests are norm-referenced, reliable and valid in diagnosing developmental motor delays [5,6]. As most developmental scales, the MABC and BOT were created in Western European (MABC-2) and North American countries (BOT-2), and item choice, population norms and cut-off scores may be limited to those regions. This raises the issue of whether these assessments would be valid in other countries [1]. Indeed, several studies have highlighted the importance of validating these assessments given the knowledge that the performance of daily life tasks may be influenced by cultural context [7]. Furthermore, the results based on the MABC in China, Czech Republic, Italy, Japan and the Netherlands suggest that there may be significant cross-regional differences in motor competence among children, and the results highlight the importance of validating these motor assessments when they are introduced to new regions [8,9,10,11,12,13].

In African countries there is a lack of motor assessments that have been validated for use in children with possible motor delays. We know of no studies that evaluated if the international age-normative values are applicable to school-aged children from African countries. In low resourced areas, for instance, having qualified physical education (PE) teachers and structured PE classes is not common [14,15,16]. Breaks are mostly used for active play and games played usually do not require (special) equipment (small shells, a rope, a piece of chalk, a ball made of plastic bags). Children with coordination difficulties have reduced levels of active play participation, which places them at risk for secondary health problems (i.e., low cardiorespiratory fitness and overweight) [17,18]. On the other hand, many children must use active transportation to school and are used to helping out parents with chores that may involve carrying and walking. Moreover, part of the children in African countries are treated for HIV with Antiretroviral Therapy, which is known to influence motor skill development, especially gross motor skills [19,20,21]. North-West province has an estimated of 26,790 children under the age of 15 living with HIV as of midyear 2016 [22], 45.4% of these children received ART. Given these child and environment related factors the children in various African countries will have developed different motor skills.

Since the psychometric properties of any measure are not universal, but rather specific to the population being reported on [23], they deserve attention when standardized tests are introduced to new world regions. Especially, establishing cross-cultural validity is of utmost importance as it may hamper the case finding of children with motor problems in different cultures, if found insufficient. As African children are likely to have developed other skills than those being originally assessed with the MABC and BOT, paying attention to and addressing these concerns about content validity [24], and reference norms is needed in order to determine whether these tests could serve as measurement tools for identifying motor problems in African children. Content validity represents the degree to which an assessment tool measures the construct it intends to measure. The items of a motor test are thought to be related to the skills children will use (and need) in their daily life (content validity) [25]. The equipment and recreational opportunities available, the educational systems (fine and gross motor training) and beliefs in the society play an important role in what skills children need in everyday activities and will acquire through natural exposure [26,27,28]. Being able to put pegs in tiny holes may be as distant a task to some children as is carrying a basket of vegetables on the head or eating with chop stick to others. Hence, motor development is largely dependent upon the experience-based learning and will therefore be context specific [29,30,31,32].

Both the MABC-2 and BOT-2 are norm-referenced, reliable and valid in evaluating motor competence and identifying motor developmental delays [5,6] and are therefore hypothesized to be related. Since psychometric properties of any measure are not universal but specific to the population, we want to explore the degree to which these two measures yield similar results (convergent validity) [33]. By determining convergent validity between the two most widely used motor skill assessments, insights are gained into whether these motor skill tests are measuring similar motor skill constructs in children raised under different environmental constraints compared to the countries where the tests were originally developed. Agreement between tests is explored when pre-established cut-points are used to identify cases at risk of motor performance development impairments such as in DCD.

The first aim in establishing cross-cultural validity of the MABC-2 and BOT-2 was to describe the classification on both tests of randomly selected school-aged South African children aged 6–10 years.

Two sub-questions were posed:(1)What are the percentages of the MABC-2 and BOT-2 classification of “at risk” and “impaired categories” for the total test and component scores in a large group of randomly selected school-aged South African children aged 6–10 years?(2)Which items cause classification into the at risk and impaired categories.

The second aim in establishing cross-cultural validity of the MABC-2 and BOT-2 was to investigate the convergent validity between both tests when completed by randomly selected school-aged children aged 6–10 years.

Five sub-questions were posed:(1)Does the BOT-2-SF standard score significantly correlate with the MABC-2 total test score when completed by children aged 6–10 years?(2)Which proportion of children are classified as having motor problems on both tests and on which percentage of children do they disagree (sensitivity/specificity)?(3)Do the BOT-2 subtest scales Running and Agility, Balance, and Strength significantly correlate with the MABC-2 total score and component scores?(4)What are the overall skill levels on sub tests of the BOT-2 of children designated as at risk for motor problems and definite motor impairments based on the MABC-2 classification.(5)Are there differences in demographics (age, sex, BMI) between children identified by both tests to have typical development, those identified by one test only, and those who do meet criteria for possible motor deficits based on both tests.

## 2. Methods

### 2.1. Participants and Procedures

In this study we included a random sample of 6- to 10-year-old children from grade 1, 2 and 3 from four general primary schools in South Africa situated in neighborhoods with a low- to low-middle socio-economic status (SES).

All children in the selected grades participated in this cross-sectional study after their parents provided written informed consent and they provided written assent themselves. The study protocol was approved by the local ethical committees of North-West University and the University of Cape Town (NWU-00491-19-A1, HREC Ref 598/2019). The parent(s) filled in the child physical activity readiness questionnaire PAR-Q [34]. Children were excluded from the sample if they had: (i) a formal diagnosis that would significantly impede motor performance as reported by the parents, (ii) refused testing, or (iii) incomplete test results due to absence from school during test administration.

Children were individually assessed with MABC-2 and BOT-2-SF in a quiet room at school. Examiners were a postgraduate pediatric physical therapist and postgraduate students with a degree in Human Movement Science specializing in Kinderkinetics (also known as a pediatric exercise scientist), who had been using the MABC-2 and BOT-2 in clinical settings regularly before the study was carried out. To avoid excess in testing time each child was tested on part of the specific subtests of the BOT-2 (see Figure 1).

#### 2.1.1. Participants Part 1 (MABC-2)

A total of 444 randomly selected children participated in the study (210/47.3% girls and 234/52.7% boys).

#### 2.1.2. Participants Part 2 (MABC-2 and BOT-2)

A subsample also performed the BOT-2-SF (14 items *n* = 165). Moreover, all items of specific subtests were also performed by part of children; Running and agility (5 items *n* = 296), Strength (5 items *n* = 161) and Balance (9 items *n* = 132).

### 2.2. Measures

The level of motor competence was assessed by the MABC-2 and BOT-2.

#### 2.2.1. The Movement Assessment Battery for Children, Second Edition (MABC-2)

The MABC-2 [3] is a normative test that measures motor skills in children from three to 16 years of age. The MABC-2 is a reliable and valid test [35]. The MABC-2 contains 8 items, divided into three components: Manual Dexterity, Aiming and Catching and Balance. The score per item, sub-score and total score can be recoded into a total standard score (SS) which considers age (range 1–19; mean score = 10; SD = 3). The raw scores from the eight different MABC-2 items were transformed, into item standard scores (ISS), component standard score (CSS) and total standard scores (TSS). On the basis of the SS each child was categorized into one of three movement difficulty categories. A standard score > 7 is regarded average/normal motor performance, 6–7 is indicative of at risk for motor problems whereas a score at or below the 5th standard score is indicative of a significant motor problem. The original UK normative data were applied for interpreting the children’s performances and setting the cut-off points [3].

#### 2.2.2. Bruininks-Oseretsky Test of Motor Proficiency, Second Edition (BOT-2)

The BOT-2 is a normative test that measures motor skills in children from four to 21 years of age. The BOT-2 is a valid and reliable assessment tool [4]. The children in this study were tested using the short form of the BOT-2. The BOT-2-SF contains 14 items taken as representative samples for each sub-test from the complete form. In the BOT-2, participants receive a raw score, which is transformed to a point score. This point score is further transformed to a scale score for the subtests or standard scores for the total score of the BOT-2-SF. The average age-adjusted scale scores for subtests are 15 (SD = 5). The age-adjusted standard scores derived for the total score of the SF range between 20 and 80 and have a mean of 50 (SD = 10).

For the specific subtests scale scores (range 1–30) were derived from point scores (Balance: Score points 0–37; Running Speed and Agility: Score points 0–52 and Strength: Score points 0–42).

On the basis of the standard scores and scale scores each child was categorized 85–100th percentile “Well above and above average”, between 18–84th percentile “Average”, between 3–17th as “Below average and impaired (<3th) [4]. For the classification comparison we combined “Well above and above average” and “Average”, to “Normal range”. The cut-off points from the original USA manual were used.

### 2.3. Data Analysis

Statistical analyses were performed with SPSS 28.0 for Mac. Demographic and classification data of MABC-2 and BOT-SF, are presented as frequency, mean +/− one Standard Deviation (SD). In order to adequately compare the results of the MABC-2 and BOT-SF, we used the standard scores for both tests, which represent normalized values, thereby considering the participants’ age.

We assessed the convergent validity between the MABC-2 and BOT-2 scores using Spearman’s correlation coefficients, as the data were not normally distributed. In order to determine the agreement in the classification for the BOT-2-SF and MABC-2 classification the children were divided in two groups on each test separately: (a) children with a score at or below the 16–17th percentile and (b) children with a score above the 16–17th percentile. The agreement between the two tests in this dichotomy was evaluated with Cohen’s Kappa (κ). According to Landis and Koch [36], a Cohen’s Kappa between 0.21 and 0.40 is considered fair, between 0.41 and 0.60 moderate, between 0.61 and 0.80 substantial and Cohen’s Kappa bigger than 0.81 is considered an almost perfect agreement.

Sensitivity and specificity of the motor scores were calculated at cut-offs using the MABC-2 as the reference standard. The usual requirement for screening tests is a sensitivity of at least 80% and a specificity of at least 90%.

Three motor performance classification groups were distinguished; children with typical development on both tests, those identified by one test only, and those who do meet criteria for motor deficits based on both tests. Differences in demographics between motor performance classification groups were examined using ANOVA (age) and χ² (distribution of Sex and BMI classification). Significance levels were set at 0.05.

## 3. Results

### 3.1. Participant Characteristics

In total 444 children participated in part 1 of the study and obtained a score mean MABC-2 total standard score of 8.40 (SD 2.74). For 5 children an item of the MABC-2 was missing so no total score could be calculated. In part 2 of the study 165 children were also tested on the BOT-2 and obtained a mean BOT-SF total standard score of 48.54 (SD 7.42). For description of demographics of the children in part 1 and 2 of the study see Table 1. Test scores on total and component scores are depicted in Table 2.

### 3.2. Classification Based on the MABC-2

In this group of 444 randomly selected children, a prevalence of 36% was found for children at risk for motor problems. As shown in Table 3, 43.2%, 22.6% and 26.7% did not meet the requirements for Manual dexterity (mean (SD) 8.3 (3.2)); Aiming and Catching (mean (SD) 9.0 (3.1)) and Balance (mean (SD) 9.2 (3.1), respectively. If we examine the data on item level, it can be noticed that half the children failed on the bicycle trial item, of which two thirds did really poor. A quarter of the children scored below the 5th percentile on the item dynamic balance (walking on a line), while children performed relatively better on static balance (one leg stance). See Table 3.

### 3.3. Classification Based on BOT-2-SF and BOT-Subtests

Percentages of children suspected of poor motor performance was about 17% based on the BOT-SF. Only 1% of the children scored below the 2nd percentile cut-off on the total score of the BOT-SF. Most children (90%), scored in the normal range on Balance (mean (SD) 15.7 (4.2)) and Running and Agility (mean (SD) 17.6 (3.5)). Items of the Strength subtest (Standing long jump, Knee push-ups, Sit-ups, Wall sit and V-up) were harder for this group of children with 78% in the normal range (mean (SD) 13.3 (3.4)). See Table 4.

### 3.4. Convergent Validity

Correlations MABC-2 and BOT-2. Table 5 lists the correlations between the Total and sub test scores of the MABC-2 and BOT-2. All correlations were significant, except between MABC-2 Aiming and catching and BOT-2 Strength. Moderate correlation (r = 0.44) was found between the two total test scores. In addition, the correlation between the two subtests intended to measure the same construct Balance was moderate (r = 0.42). Interestingly the highest correlation (r = 0.46) was found between MABC-2 Total Test Score (TTS) and the BOT-2 subtest Running and Agility. Strength on the other hand is the least associated with MABC-2 outcomes.

### 3.5. Agreement between Classifications

#### Sensitivity Specificity

In total 68% (108/160) of the children were classified comparably on the 2 tests (91 as normal range and 17 as at risk). Of the children classified by the MABC-2 as at risk, 32% (*n* = 42) scored in the normal range of the BOT-2. Of the children classified by the BOT-2 as at risk 37% (*n* = 10) scored in the normal range of the MABC-2 (Kappa 0.213, *p* = 0.002). See Table 6 and Figure 2. If we use the MABC-2 as gold standard, the BOT-2 has a sensitivity of 0.29 (CI 0.19–0.41) and a specificity of 0.90 (CI 0.83–0.95). Indicating 71% false negative results (Type II error) and 10% false positives (Type I errors).

### 3.6. Differences in Demographic Characteristics between Motor Performance Classification Groups

Children who were classified below the 16th percentile on both tests, on one test or in the normal range on both tests, were not different in age (*p* = 0.48) and gender (*p* = 0.32) distribution. However, BMI classification was different between groups (χ² (12.66), *p* 0.049). More children who were classified below the 16th percentile on both tests were obese. This was mainly caused by the BOT-2 classification. Obesity was found in 8.1% of children in the normal range and in 22.1% of children in the at-risk category of the BOT-2, while obesity was shown in 9.6% of the normal range children and 11.6% of the children in the at-risk category of the MABC-2.

## 4. Discussion

This study examined cross-cultural validity of the MABC-2 and BOT-2 in a randomly selected sample of school-aged South African children. Based on the outcomes of the MABC-2 a prevalence of 36% was found for children at risk for motor problems. Almost 20% of the children obtained scores in the impaired category for the component Manual dexterity. Based on BOT-SF outcomes the percentages of children suspected of poor motor performance was about 17%. The subtest of the BOT-2 with the lowest scores was Strength, with 22% of the children in the at-risk category. The BOT-2-SF total standard score and MABC-2 total test score showed a moderate correlation (r = 0.44) and a fair Kappa of 0.21. Since we used a random group of sufficient sample size, we were able to examine important measures of agreement such as sensitivity and specificity. The specificity of 0.90 tells us how likely it is for the BOT-2 to be negative (no motor difficulties) in case someone does not have motor difficulties. The sensitivity of 0.21 tells us that many children who should have tested positive for motor difficulties based on MABC-2 outcome did not do so on the BOT-2. Lastly there were differences in demographics between motor performance classification groups; children who scored positive (at risk or impaired) on both tests were more often obese than children with other combinations.

The fact that 36% of the children without known disorders scored at or below the 16th percentile of the MABC-2 could mean two things: (1) one third of the children in the current sample has poor motor skills, or (2) the international age-normative mean values are not likely to be applicable to children from African countries. MABC-2 is an easy instrument for screening children over a short period of time and the manual has been translated into numerous languages. However, two points are important to consider. First, generalization of the validity findings is limited to populations with similar attributes, and in similar contexts, as those in whom the instrument was tested for validity. If that is not checked, these instruments may lack to provide information relevant to a specific population living under specific/different circumstances and may be more sensitive to exploration (experience) differences than to motor impairment. Second, even if the instrument is valid, it could still be that norms (and cut-off values) from the original population are not suited for children living in the different context. On the other hand, using the same tool and cut-off scores for research has advantages to compare data across demographic or clinical populations.

### 4.1. Prevalence of Poor Motor Skills

The percentage of children scoring in the “at risk for motor problems” range was 36% which is much higher than the current estimated prevalence in school-aged children [37]. Children in different countries will perform different daily tasks, each with different facilitators and barriers in their environments [28,38,39]. DCD has been reported to be more prevalent in low-resourced areas, therefore, differences in percentage are to be expected [40]. Children in the current sample live in the settings where they often experience limitations with equipment, coaching and space as barriers to participate in physical activities, such as sports. This in turn limits their development of coordinated motor skills [40]. In particular, art and crafts, and handwriting training is less rigorous due to space (more children in one bench with rough surface or floor seating) and material limitations. Although this information was not sampled, part of the children in the current sample were treated for HIV with Antiretroviral Therapy which might also have influenced their motor skill development. In summary there are reasons to belief that the level of skills on the items included in the MABC-2 might indeed be lower in the random South African sample than in the UK sample.

### 4.2. BOT-2 MABC-2 Comparison

Although at first sight there are many similarities between MABC-2 and BOT-SF in that they are both focusing on functional motor skills (e.g., fine motor and gross motor skills and balance) and yield a quantitative measure that indicates performance level (norm-referenced values), correlations between the scores (r = 0.44) indicate that they partly measure different constructs or required levels of the motor skills. Thus, these two tests do not meet the criterion for testing similar constructs: (r > 0.5) but of related constructs (0.30–0.50) [25].

Due to the different nature of motor performance assessment tools, it is not uncommon that different tests identify different children. The number of gross and fine motor items vary between the two tools. Even comparable items have small differences that may lead to different results. For instance, the BOT-2 balance items only last 10 s while the MABC-2 have a maximum time of 30 s. On the other hand, the BOT-2 subtest Balance also includes three items with eyes closed. The time difference may select children with different underlying problems, 30 s is harder but also more prone to pick up distractibility, while the eyes closed items may be more sensitive to poor proprioception. The MABC-2 categorizes jumping or hopping as balance while the BOT-2 orders these items under the construct Running and Agility. Moreover, no items for Bilateral coordination and Strength are included in the MABC-2.

So far, we found only two studies comparing the BOT-2 and MABC-2 in this age range [41,42]. Other studies either used an earlier version of the tests [43,44] or looked at specific groups (intellectual disabilities: [45]). The correlation between the first version of the MABC and BOT was also moderate (r = 0.50, *p* < 0.01). Kappa’s were also low (k = 0.19 at the 5th percentile; k = 0.29 at 15th percentile cut-points), so comparable to our study [46]. Spironello and colleagues suggested that the BOT rather than the M-ABC may be preferable in school-based studies where large numbers of children need to be assessed at the same time and where cost prohibits the administration of motor tests by health professionals [46].

Of the studies comparing second edition of both tests, one study was performed as part of the validation for the Dutch version of the MABC-2. In this study, we compared the full-scale BOT-2 to MABC-2 and a correlation of 0.58 (*p* < 0.01) was found [42]. Although we commonly speak about “the” MABC-2, this test in fact constitutes 3 different test sets, which measure different constructs per age group. This was confirmed when Lane and Brown [41] examined the convergent validity of the MABC-2 and BOT-2 in 25 children in age band 2 and 25 in age band 3. Importantly, no significant correlation was found between the BOT-2 Total Motor Composite and Total Test Score of the MABC-2 in the age band 2. However, a positive strong correlation was found between the BOT-2 Total Motor Composite and MABC-2 Total Test Score in the age band 3 (rho = 0.80, *p* < 0.01).

While we cannot rule out the option that the international age-normative values of the MABC-2 (age band 2) are not applicable to children from African countries, after comparing the results in our study to the outcomes of the BOT-2, it seems the less obvious explanation. Thus far, the prevalence in the current sample of the children seems within the estimated for motor problems on the BOT-2, around 17 percent. The BOT-2 subtests showed no clear picture of high prevalence of low levels of Running and Agility and Balance. For the sub test Strength, the results were less favorable (22% in the lower range), which could be caused by real strength deficits but also partly by the children not being used to making isolated movements. These movements may be more familiar to children doing physical education classes (push-ups and sit-ups) tested in the American norm sample.

### 4.3. Differences in BMI between Motor Performance Classification Groups

Obesity in the current sample was 10 percent (*n* = 17), seven of these children showed poor motor skill levels on both tests. Although one needs to be cautious given these small numbers, the finding that children with low motor coordination had higher BMI concurs with the literature. Hendrix and colleagues, based on a systematic review, concluded that the prevalence of overweight and obesity was consistently higher in children with DCD in studies originating in Australia, Canada, Greece, the Netherlands, Taiwan and Hong Kong [17]. De Meester et al. also reported that 90% of children in a cohort with low motor coordination did not meet the daily physical activity recommendations for children [47]. Along with unhealthy dietary habits, physical inactivity is an important risk factor for developing overweight and obesity [48]. It was suggested by Rivilis that the detrimental effect of poor coordination on body composition might not manifest itself until later in childhood or early adolescence [18]. However, our research was on younger children (6–10 year), most of which are on a food program. This makes the food intake between our participants with different levels of motor competency more similar, compared to children studied in Western countries. Our results make the explanation that children with low motor skills are more likely to avoid participation in physical activities leading to a secondary exercise deficit disorder and weight gain, feasible [26,49].

## 5. Conclusions

Far more children scored in the clinical “at risk” range (<16th percentile) when tested with the MABC-2 than with the BOT-2-SF. Overall, these children seemed not to be limited in motor performance measured by the BOT-2-SF, Running and Agility, and Balance. However, South African children did show lower levels of strength. A preliminairy conclusion is that the items and cut off value of MABC-2 are less suited for the groups tested in this study. Children from different cultures will need tests for the specific motor skills that are required for optimal functioning in their own setting. Motor skill test items for typical daily activities in these areas may require more traditional physical activities that children use on a regular basis to measure their motor skill efficiency.

## Figures and Tables

**Figure 1 ijerph-19-06788-f001:**
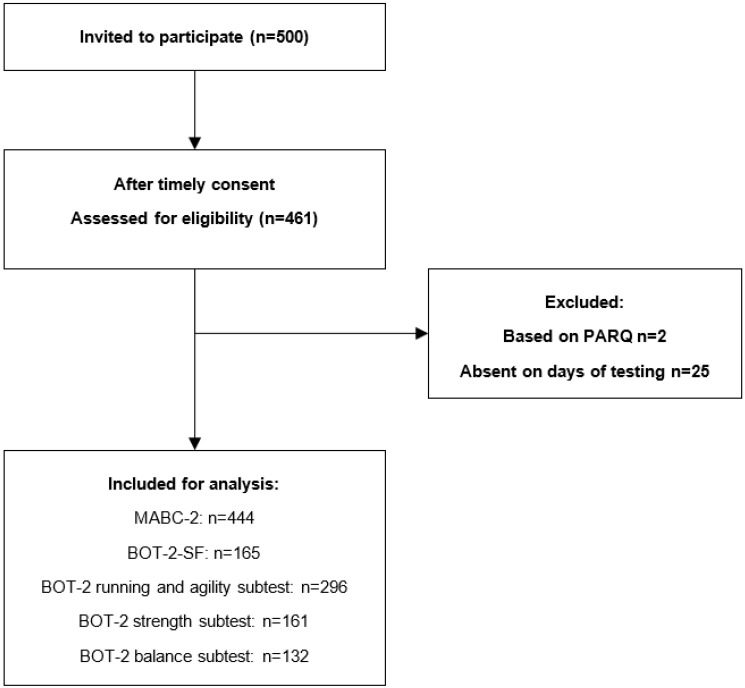
Flowchart of the selection procedure and number of children in the final analysis. Children whose caregivers answered in the affirmative on any of the questions of the children’s Physical Activity Readiness Questionnaire (PAR-Q) were excluded from the study.

**Figure 2 ijerph-19-06788-f002:**
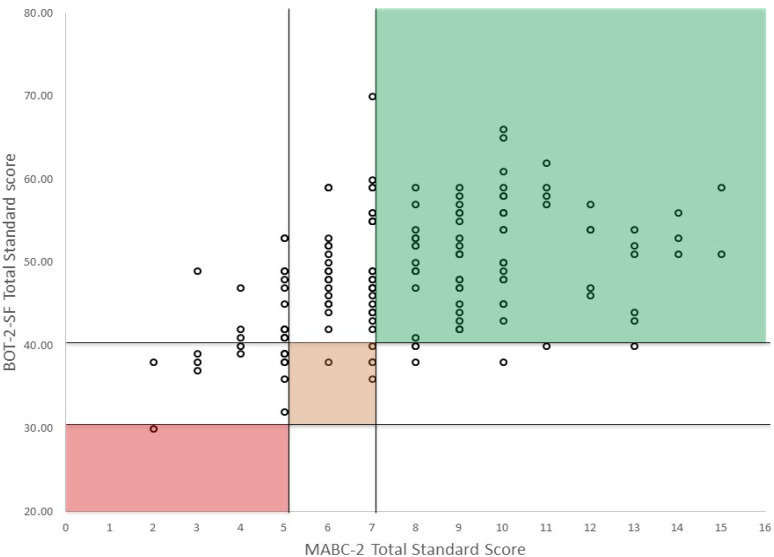
Classification agreement between MABC-2 and BOT-2 (Green: no movement difficulty, Amber: at risk of having movement difficulty; Red: significant movement difficulty). Colored areas show children that had the same classification of the two tests. Please note the children on the left side of the graph (0–5 standard scores on the MABC-2: significant motor difficulty) who score in the normal range (>40) on the BOT-2.

**Table 1 ijerph-19-06788-t001:** Demographic information of Participants Part 1 (MABC-2; *n* = 444) and Participants Part 2 (MABC-2 and BOT-2; *n* = 165).

	Groups	
	MABC-2	MABC-2 + BOT-2
Boys/girls (%)	52.7/47.3	53.9/46.1%
Age (years, mean (SD))	7.7 (1.0)	7.9 (0.96)
Height (cm, mean (SD))	128.4 (8.89)	128.0 (8.66)
Weight (kg, mean (SD))	28.4 (8.77)	28.8 (9.29)
BMI (kg/m^2^, mean (SD))	16.95 (3.5)	17.28 (3.7)

Legend: MABC-2: Movement Assessment Battery for Children, 2nd edition; BOT-2: Bruininks-Oseretsky for Motor Proficiency, 2nd edition.

**Table 2 ijerph-19-06788-t002:** Test scores of the sample and subsamples.

	N	Min	Max	Mean (SD)
**MABC-2 (standard score)**				
▪ Components				
○ Manual dexterity	442	1	19	8.3 (3.2)
○ Aiming & Catching	442	1	16	9.0 (3.1)
○ Balance	439	1	16	9.2 (3.1)
▪ Total score	439	2	16	8.4 (2.7)
**BOT-2 (standard score)**				
▪ Specific subtests				
○ Balance subtest	132	7	26	15.7 (4.2)
○ Running & agility subtest	296	5	26	17.6 (3.5)
○ Strength subtest	161	0	23	13.3 (3.4)
▪ Short form (total)	165	30	70	48.5 (7.4)

Legend: MABC-2: Movement Assessment Battery for Children, 2nd edition; BOT-2: Bruininks-Oseretsky for Motor Proficiency, 2nd edition; N: number of children; Min: Minimum; Max: Maximum.

**Table 3 ijerph-19-06788-t003:** Percentage per motor skill classification based on MABC-2 total scores, component and item scores.

Expected %	84%	11%	5%
	Normal	At Risk	Impaired
**Total Score MABC-2**	**64.0**	**19.6**	**16.4**
**Components**			
▪ Manual Dexterity	56.8	23.5	19.7
▪ Aiming and Catching	77.4	6.1	16.5
▪ Balance	73.3	15.7	10.9
**Items**			
▪ Item1 One hand	69.1	17.6	13.3
▪ Item2 Bi manual	72.1	12.2	15.8
▪ Item3 Bicycle trail	50.0	12.4	37.6
▪ Item4 Throw	74.5	10.4	14.7
▪ Item5 Aiming	63.3	18.1	18.6
▪ Item6 Static Balance	70.0	21.1	8.2
▪ Item7 Slow Dynamic Balance	67.1	6.8	25.6
▪ Item8 Jump/Hop	79.4	9.5	11.1

Legend: MABC-2: Movement Assessment Battery for Children, 2nd edition.

**Table 4 ijerph-19-06788-t004:** Percentage per motor skill classification based on BOT-2 total scores and subtest scores.

Expected %	83%	15%	2%
	Normal	At Risk	Impaired
**Total BOT-2-SF**	**84**	**16**	**1**
Subtests			
▪ Balance	91	9	0
▪ Running and Agility	97	2	0
▪ Strength	78	21	1

Legend: BOT-2-SF: Bruininks-Oseretsky for Motor Proficiency, 2nd edition, Short Form.

**Table 5 ijerph-19-06788-t005:** Spearman’s rho correlation coefficients between the MABC-2 standard scores and the BOT-2-SF age adapted scaled scores.

		BOT-2-SF (SS)							
		*Total Score*		*Subtests*					
				Balance		Running & Agility		Strength	
MABC-2		rho	*p*-value	rho	*p*-value	rho	*p*-value	rho	*p*-value
**Total score (SS)**		**0.439**	<0.001	**0.411**	<0.001	**0.458**	<0.001	**0.239**	0.003
**Components**	Manual Dexterity	**0.322**	<0.001	**0.211**	0.015	**0.378**	<0.001	0.168	0.035
	Aiming & Catching	**0.218**	0.005	**0.349**	<0.001	**0.197**	0.001	0.119	0.136
	Balance	**0.409**	<0.001	**0.421**	<0.001	**0.361**	<0.001	**0.239**	0.003

Legend: MABC-2: Movement Assessment Battery for Children, 2nd edition; BOT-2-SF: Bruininks-Oseretsky for Motor Proficiency, 2nd edition, Short Form. Significant values are printed bold.

**Table 6 ijerph-19-06788-t006:** Sensitivity and specificity of MABC-2 and BOT-2 for 160 children with complete results. Sensitivity: 0.29 (CI: 0.18 to 0.41) Specificity: 0.90 (CI: 0.83 to 0.95).

			MABC-2		Total
			Below 16th Percentile	Above 16th Percentile	
**BOT-2**	Below 17th percentile	Number (n)	17	10	27
		% within BOT-2	63%	37%	100%
		% within MABC-2	28.8%	9.9%	16.9%
	Above 17th percentile	Number (n)	42	91	133
		% within BOT-2	31.6%	68.4%	100%
		% within MABC-2	71.2%	*90.1%*	83.1%
**Total**		Number (n)	59	101	160
		% within BOT-2	36.9%	63.1%	100%
		% within MABC-2	100%	100%	100%

## Data Availability

The data presented in this study are available on request from the corresponding author. The data are not publicly available due to privacy.

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
