# Peer review of "Exploring Cultural Bias in Two Different Motor Competence Test Batteries When Used in African Children"

_ijerph, 2022, doi:10.3390/ijerph19116788_

Round 1

Reviewer 1 Report

In the discussion everything indicated is rather to respond to the results since what is indicated here are characteristics and description of the tests applied, I suggest including it in the results and continuing the discussion with the following background information that truly responds to the discussion in the section 4.1.

The food variable is not considered during the study and if there is concern about giving an answer to the nutritional aspect addressed from the BMI, it should go in a separate chapter, such as complementary comments within the section on difficulties for the study. Section 4.3

Author Response

Comment: In the discussion everything indicated is rather to respond to the results since what is indicated here are characteristics and description of the tests applied, I suggest including it in the results and continuing the discussion with the following background information that truly responds to the discussion in the section 4.1.

Answer: We appreciate the comments of the reviewer, however current practice is to begin with a brief restatement of the main findings. We agree that there is some repetition in the discussion but we belief it really helps the reader to have the most important finding mentioned when it is discussed (to avoid that people have to go back to the tables for each point in the discussion). We tried to cut down a few sentences in the discussion to shorten the tekst.

Comment: The food variable is not considered during the study and if there is concern about giving an answer to the nutritional aspect addressed from the BMI, it should go in a separate chapter, such as complementary comments within the section on difficulties for the study. Section 4.3.

Answer: We thank the reviewer for this point. (S)He is right, we did not consider food during the study. But as is standard we described weight and height (BMI ) as descriptive background info. The literature reports higher levels of BMI in children with neurodevelopment disorders. We found it unexpected that this was also the case in our most severely affected children. We added info about relation between weight and motor skill in the introduction.  

Reviewer 2 Report

The proposed article is a case study, applying two useful tools to a population that has not been analyzed. The results highlight a number of conclusions that may provide the opportunity to start other studies that bring more information.

The article meets the standards of a research. however, a few small details can add value:

- I suggest that the punctuation mark in the title be removed

- I suggest including more keywords (eg African children, etc.)

- reviewing the presentation of the tools used (many details are sometimes lost to the reader), but also of the results (correlating the description of the results with the included tables - for example, see the information related to tables 2 and 5, but also in subchapter 3.6)

- identification of more recent studies (some are older than 15 years)

- the subchapter Participants should be revised: it is not very clear how many of the 444 participated in both tests, taking into account the data for the 3 variables invoked: gender, age, BMI - I suggest including the information in a table. also to analyze the effects of their interaction on the analyzed dependent variables.

- also in chapter 4.1. Prevalence of poor motor skills, reference is made to a variable for which data are not included in previous chapters (at least for Participants): children with HIV - ”part of the children in the current sample were treated for HIV with Antiretroviral Therapy which is known to influence motor skill development ”. I suggest reconsidering the way the information is presented.

Author Response

Reviewer 2

The proposed article is a case study, applying two useful tools to a population that has not been analyzed. The results highlight a number of conclusions that may provide the opportunity to start other studies that bring more information.

Answer: We thank to reviewer for taking the time to look at our paper and appreciate the precise comments given to improve the paper.

The article meets the standards of a research. however, a few small details can add value:

- I suggest that the punctuation mark in the title be removed.

Answer: Thank you for this suggestion the punctuation mark in the title has been removed.

- I suggest including more keywords (eg African children, etc.)

Answer: We added African to children in the keywords although if a word is in the title it will come up in a search.

- reviewing the presentation of the tools used (many details are sometimes lost to the reader), but also of the results (correlating the description of the results with the included tables - for example, see the information related to tables 2 and 5, but also in subchapter 3.6)

Answer: We added a reference to Table 2 in subchapter 3.1 ‘Test scores on total and component scores are depicted in table 2’. We have added more detail in subchapter 3.6 and hope it is more clear now.

- identification of more recent studies (some are older than 15 years).

Answer: We thank the reviewer for this comment. We added more recent studies to the reference list:

The international DCD guideline in which the MABC-2 and BOT-2 are recommended for use in children with DCD: Blank, R.; Barnett, A.L.; Cairney, J.; Green, D.; Kirby, A.; Polatajko, H.; Rosenblum, S.; Smits-Engelsman, B.; Sugden, D.; Wilson, P.; et al. International clinical practice recommendations on the definition, diagnosis, assessment, intervention, and psychosocial aspects of developmental coordination disorder. Dev Med Child Neurol 2019, 61, 242-285, doi:10.1111/dmcn.14132.

Recent studies reporting on the impact of cultural and/or environmental factors on motor competence and/or ADL in children.

  • True, L.; Pfeiffer, K.A.; Dowda, M.; Williams, H.G.; Brown, W.H.; O'Neill, J.R.; Pate, R.R. Motor competence and characteristics within the preschool environment. J Sci Med Sport 2017, 20, 751-755, doi:10.1016/j.jsams.2016.11.019.
  • Delgado-Lobete, L.; Montes-Montes, R.; Pértega-Díaz, S.; Santos-Del-Riego, S.; Cruz-Valiño, J.M.; Schoemaker, M.M. Interrelation of Individual, Country and Activity Constraints in Motor Activities of Daily Living among Typically Developing Children: A Cross-sectional Comparison of Spanish and Dutch Populations. Int J Environ Res Public Health 2020, 17, doi:10.3390/ijerph17051705.
  • Adolph, K.E.; Hoch, J.E. Motor Development: Embodied, Embedded, Enculturated, and Enabling. Annu Rev Psychol 2019, 70, 141-164, doi:10.1146/annurev-psych-010418-102836.
  • Hallemans, A.; Verbeque, E.; Van de Walle, P. Motor functions. Handb Clin Neurol 2020, 173, 157-170, doi:10.1016/b978-0-444-64150-2.00015-0.
  • van der Linde, B.W.; van Netten, J.J.; Otten, E.; Postema, K.; Geuze, R.H.; Schoemaker, M.M. A systematic review of instruments for assessment of capacity in activities of daily living in children with developmental co-ordination disorder. Child Care Health Dev 2015, 41, 23-34, doi:10.1111/cch.12124.
  • Krieger, B.; Schulze, C.; Boyd, J.; Amann, R.; Piškur, B.; Beurskens, A.; Teplicky, R.; Moser, A. Cross-cultural adaptation of the Participation and Environment Measure for Children and Youth (PEM-CY) into German: a qualitative study in three countries. BMC Pediatr 2020, 20, 492, doi:10.1186/s12887-020-02343-y.

- the subchapter Participants should be revised: it is not very clear how many of the 444 participated in both tests, taking into account the data for the 3 variables invoked: gender, age, BMI - I suggest including the information in a table. also to analyze the effects of their interaction on the analyzed dependent variables.

Answer: The reviewer may have missed that detailed information about the number of children in each test. They are presented in the flow chart (figure 1) and in table 1 (now table 2). As requested we have put the demographic information in a new table (now table 1).

- also in chapter 4.1. Prevalence of poor motor skills, reference is made to a variable for which data are not included in previous chapters (at least for Participants): children with HIV Comment:  ”part of the children in the current sample were treated for HIV with Antiretroviral Therapy which is known to influence motor skill development ”. I suggest reconsidering the way the information is presented.

Answer: We added general information about effect of Antiretroviral Therapy to the introduction. We have no extra information on this topic (given the privacy) but still feel it is important to mention it as a possible factor in motor development.

Reviewer 3 Report

First and foremost, I would like to extend my sincere congratulations to the authors for their work. The resulting paper shed light to some very interesting and enriching topics, From my point of view, the article is really interesting.

I simply think that it would be interesting modified the discussion because I think that it might be too long and the method. It could be interesting remove some parts of the text if researches think it appropriate.

Materials and methods:

The design of the method section could perhaps be structured differently in order to clarify the context.

Discussion:

It might be too long.

References:

First reference number 1, It should start with a capital letter.

Author Response

Reviewer 3

First and foremost, I would like to extend my sincere congratulations to the authors for their work. The resulting paper shed light to some very interesting and enriching topics, From my point of view, the article is really interesting.

I simply think that it would be interesting modified the discussion because I think that it might be too long and the method. It could be interesting remove some parts of the text if researches think it appropriate.

Answer: We thank the reviewer for these kind words and the acknowledgments for the hard work of doing these large-scale field studies.

Materials and methods:

The design of the method section could perhaps be structured differently in order to clarify the context.

Answer: It was hard for me to respond to this suggestion because no indication was made how to restructure. But based on the comments from the other reviewers we have made changes in the participants and results section which may also relate to this point made by the reviewer.

Discussion:

It might be too long.

Answer: We have taken some sentences out and moved some parts to the introduction.